# Anomalous non-equilibrium response in black phosphorus to sub-gap mid-infrared excitation

Angela Montanaro [1,2], Francesca Giusti[1,2], Matteo Zanfrognini[3], Paola Di Pietro[2], Filippo Glerean[1,2], Giacomo Jarc[1,2], Enrico Maria Rigoni[1,2], Shahla Y. Mathengattil[1,2], Daniele Varsano [4], Massimo Rontani [4], Andrea Perucchi[2], Elisa Molinari [3,4] & Daniele Fausti [1,2 ✉]

The competition between the electron-hole Coulomb attraction and the 3D dielectric screening dictates the optical properties of layered semiconductors. In low-dimensional materials, the equilibrium dielectric environment can be significantly altered by the ultrafast excitation of photo-carriers, leading to renormalized band gap and exciton binding energies. Recently, black phosphorus emerged as a 2D material with strongly layer-dependent electronic properties. Here, we resolve the response of bulk black phosphorus to mid-infrared pulses tuned across the band gap. We find that, while above-gap excitation leads to a broadband light-induced transparency, sub-gap pulses drive an anomalous response, peaked at the single-layer exciton resonance. With the support of DFT calculations, we tentatively ascribe this experimental evidence to a non-adiabatic modification of the screening environment. Our work heralds the non-adiabatic optical manipulation of the electronic properties of 2D materials, which is of great relevance for the engineering of versatile van der Waals materials.

[1] Department of Physics, Università degli Studi di Trieste, 34127 Trieste, Italy. [2] Elettra Sincrotrone Trieste S.C.p.A., 34127 Basovizza, Trieste, Italy. [3] Dipartimento FIM, Università degli Studi di Modena e Reggio Emilia, 41125 Modena, Italy. [4] Consiglio Nazionale delle Ricerche—Istituto Nanoscienze, 41125 Modena, Italy. ✉email: daniele.fausti@elettra.eu

Layered van der Waals (vdW) materials have received increasing attention in recent years due to their potential applications in optoelectronics and solar energy harvesting[1–3]. Due to spatial confinement, the optical response of atomically thin semiconductors is dominated by strongly bound excitons, whose typically large binding energies are drastically reduced by more than an order of magnitude in their bulk counterpart. Understanding how the excitonic structure is affected by the dimensionality crossover from 2D to 3D is of key importance to clarify the role of interlayer coupling and dielectric screening in defining the electronic properties of layered semiconductors. However, this is intrinsically challenging in most of 2D materials, like transition-metal dichalcogenides and hexagonal boron nitride semiconductors, mainly because the band gap is direct only in the monolayer, while indirect in the bulk material. Black phosphorus (BP) has recently emerged as a promising candidate to bridge this gap[4]. With a carrier mobility comparable to that of graphene, BP features a direct band gap in both its single-layer (phosphorene) and bulk form[4,5]. The amplitude of the gap is strongly layer-dependent, spanning from the visible (2 eV) to the mid-infrared (MIR) (0.3 eV) range as the layer thickness is increased, thereby making BP a unique platform to study the dimensionality crossover from 2D to 3D.

The application of external stimuli, such as electric fields[6], pressure[7,8], in-plane mechanical strain[9–12] and dopants[13,14], has proved an effective way to manipulate the electronic structure of BP. Ultrashort pulses are emerging as an exceptional tool to address the non-equilibrium response and possibly control both its electronic and optical properties on the ultrafast time-scale[15–20]. In particular, photo-excitation by near-infrared laser pulses has been found to trigger a band gap renormalization in bulk BP due to the transient enhancement of dielectric screening induced by the excited photo-carrier population[21–23]. However, the non-equilibrium dielectric environment upon photo-injection of a small (i.e., smaller than the band gap) excess energy, remains unexplored. In different complex materials, long-wavelength ac-fields have been shown to hinder electronic excitations and drive collective, non-adiabatic responses, through phonon-pump[24–27] and coherent electronic effects[28–31].

In this work, we investigate the non-equilibrium response of bulk BP to both above-gap and sub-gap photo-excitation. Experimentally, we photo-excite the sample by ultrashort pulses with photon energy tunable across the equilibrium MIR band gap and monitor the photo-induced change in reflectivity over a broad energy range (Fig. 1a, b). We find that photo-excitation by high- and low-photon energy pulses yields remarkably different optical responses. While high-photon energy excitation leads to a broadband light-induced transparency in the visible range due to phase space filling, excitation with photon energies comparable to the band gap triggers an anomalous response, which is solely localized at the energies of the single-layer exciton resonance. We characterize this response as function of the MIR photon energy, the photo-excitation intensity and the sample temperature. The appearance of a transient response peaked at the energies compatible with the single-layer exciton suggests that the response observed may be related to a dynamical modification of the 3D screening. We speculate that these changes could be associated to the non-adiabatic coherent dynamical redistribution of the charge carriers driven by MIR pulses, which may trigger an impulsive modification of the interlayer interactions and, in turns, of the 3D screening. While further studies are needed to confirm this scenario, our findings unlock a new route for the optical manipulation of 2D materials and exciton-based applications to optoelectronic switches.

## Results

**Equilibrium dielectric environment in black phosphorus**. BP is an elemental semiconductor which crystallizes in a layered orthorhombic structure, where, because of sp3 orbital hybridization, phosphorus atoms are arranged in a puckered honeycomb lattice[32]. As shown in Fig. 1a, the resulting layers have two inequivalent high-symmetry directions, the so-called armchair ($\hat{x}$) and zig-zag ($\hat{y}$) directions. This strong in-plane anisotropy has macroscopic consequences on both the electronic and optical properties of BP[5,33,34]. We show in Fig. 1c the single-particle optical absorption in bulk BP calculated through first-principle hybrid functional DFT theory (see Methods). As a result of the symmetry selection rules, light polarized along the zig-zag direction is expected to be absorbed only above ∼ 2 eV. Conversely, when the polarization is parallel to the armchair direction, the computed absorption threshold is 0.345 eV. This agrees with previous experimental and theoretical studies[5,35], and corresponds to a direct electronic band gap at the Γ point (Fig. 1b).

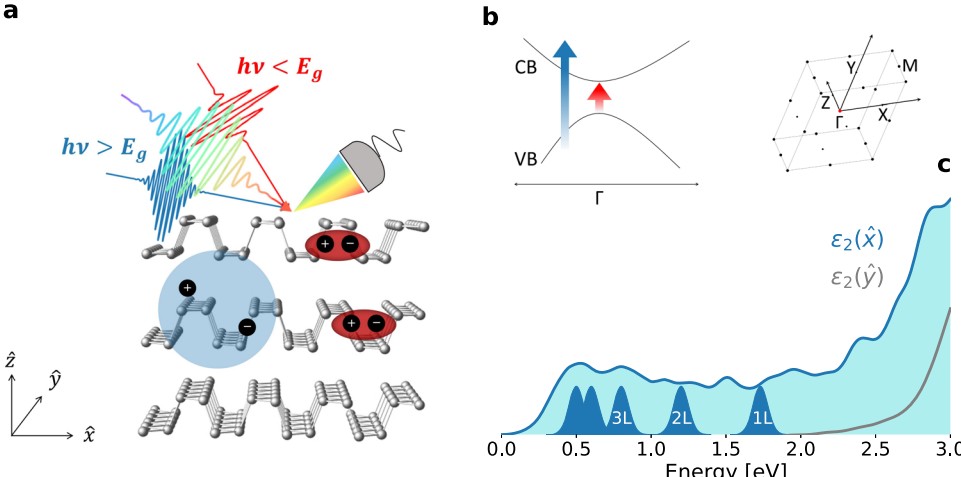

**Fig. 1 Optical fingerprint of 3D screening in bulk black phosphorus. a** Sketch of the broadband reflectivity measurements carried out on bulk BP following above- (blue) and below-gap (red) photo-excitation. Mid-infrared photo-excitation triggers an optical resonance that is consistent with the undressed exciton in the monolayer phosphorene. **b** Simplified sketch of the electronic structure (VB valence band, CB conduction band) at the Γ point, as indicated in the first Brillouin zone on the right. A conventional unit cell with double sized interlayer distance was used in the calculations, as in ref. [62]. The arrows represent the high-photon energy (blue) and sub-gap (red) photo-excitation. **c** Calculated imaginary part of the dielectric function ($\varepsilon_2$) in bulk BP along the armchair ($\hat{x}$, blue) and zig-zag ($\hat{y}$, gray) direction, convoluted with a numerical broadening of 100 meV. The blue peaks denote the lowest-energy exciton resonances ($E_{11}$) in monolayer (1 L), bilayer (2 L), trilayer (3 L) etc. BP, as measured in ref. [36].

In order to identify the optical transitions that give rise to the structured absorption in Fig. 1c, it is crucial to consider the evolution of the optical properties of BP as function of the sample thickness. The direct band gap energy in the monolayer phosphorene is $\sim 2$ eV, and the optical absorption is dominated by excitons with binding energies as large as hundreds of meV[36–40]. Although higher-energy exciton states are theoretically predicted in the single-layer limit[5], the most prominent one is the lowest-energy 1 s resonance of the $E_{11}$ exciton transition, that has been identified by photoluminescence and absorption spectroscopy, and lies at $\sim 1.73$ eV[36,41]. Importantly, as the layer number (L) is increased, the $E_{11}$ resonance (along with the band gap) monotonically shifts to lower energies, as a consequence of the strong interlayer interactions. As denoted by the position of the blue peaks in Fig. 1c, the singlet exciton red-shift does not scale linearly with the layer number and eventually reaches a plateau at $\sim 15$ L, giving rise to the absorption edge in the bulk limit (0.34 eV)[37]. This critical thickness is consistent with the theoretically predicted screening length of 10 nm, which is roughly 20 atomic layers[5]. The optical absorption of bulk BP is thus the result of the strong interlayer interactions that arise from the reduced perpendicular quantum confinement. In particular, the optical transitions within 0.34 eV (gap energy in bulk BP) and 2 eV (gap energy in the monolayer) are intrinsically related to the three-dimensional nature of the material[42]. This is evident in our DFT calculations, which confirm that absorption in this optical range is dominated by transitions involving solely the lowest-energy bands dispersing along the stacking direction, as detailed in Supplementary Note 11. In our time-domain measurements, we leverage on this characteristic and use a white-light supercontinuum (1.3–2.2 eV) to monitor the transient reflectivity in this spectral region that is expected to be the most sensitive to pump-induced changes of the 3D screening environment.

**Transient response to above- and sub-gap excitation.** The broadband transient reflectivity measurements were carried out on freshly cleaved bulk BP employing the experimental setup described in ref. [43]. From the steady-state Fourier transform infrared (FTIR) measurements reported in Supplementary Fig. 1, we identified the gap energy at room temperature to be $\sim 0.33$ eV. In contrast to most of semiconductors, the band gap in bulk BP redshifts at lower temperatures, to reach $\sim 0.28$ eV at 12 K. This anomalous behavior has been widely observed[44–47], and only recently related to the temperature-dependence of the interlayer vdW coupling[48]. We used the FTIR measurements as a benchmark to tune the photon energy of the pulses that drive the sample out of the equilibrium conditions. We investigated two distinct regimes: (1) the photo-injected excess energy is larger than the band gap ($h\nu > E_g$), so a carrier population is excited in the material; (2) the sample is photo-excited by MIR pulses ($h\nu < E_g$) that are not energetic enough to initiate electronic transitions.

We show in Fig. 2c the time- and energy-resolved transient reflectivity change upon high-photon energy excitation. The measurement reported was performed at T = 10 K, but we stress that this result is not affected by the sample temperature (Supplementary Fig. 5). For probe photon energy ($E_{pr}$) below 2 eV, the pump-probe time traces are characterized by an initial negative change in reflectivity, followed—after $\sim 1$ ps—by a less intense positive signal (Fig. 2d), that lasts for hundreds of ps (Supplementary Fig. 2). These results are consistent with previous quasi-monochromatic pump-probe experiments in the near-infrared range, in which the early-time signal is ascribed to photo-bleaching due to Pauli blocking and the subsequent one to photo-induced absorption by the excited free-carrier population[15,16,18].

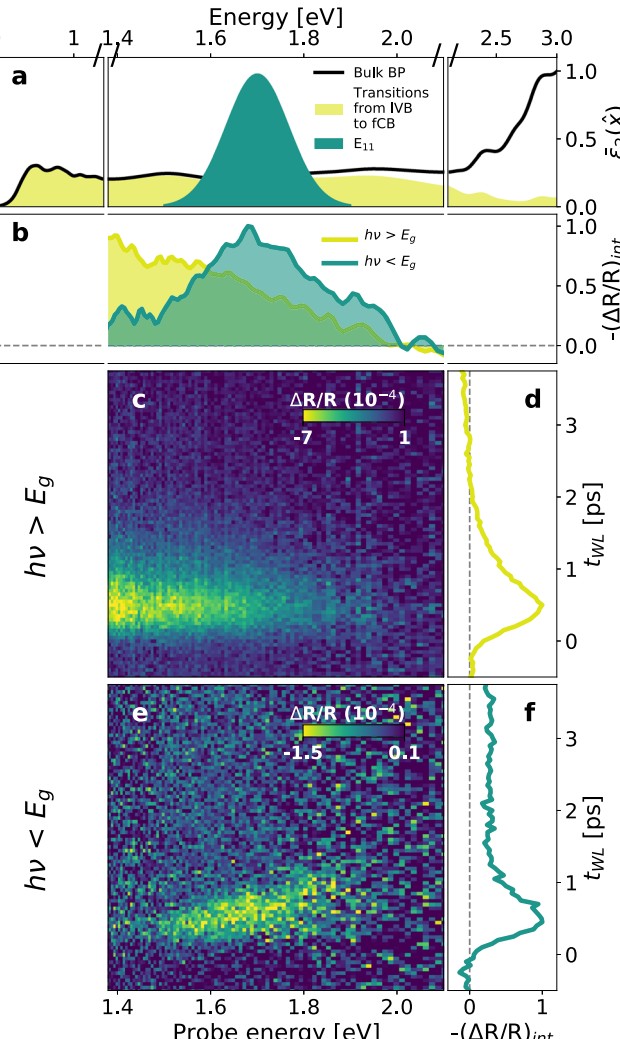

**Fig. 2 Above- and below-gap photo-excitation in bulk black phosphorus. a** Normalized DFT-calculated total optical absorption of bulk BP (thick black line, $\bar{\varepsilon}_2$). The yellow area indicates the optical absorption obtained when only the transitions from the last valence band (lVB) to the first conduction band (fCB) are included in the calculation. The green Gaussian shape indicates the lowest-energy exciton resonance in the single-layer limit ($E_{11}$). The energy scale is uniformly spaced from 1.34 eV up to 2.1 eV (probe energy window), and it is shrinked below 1.38 eV and above 2.1 eV. **c, e** Transient reflectivity map measured on bulk BP at 10 K upon photo-excitation by high-photon energy (3.1 eV $> E_g$) and sub-gap (275 meV $< E_g$) pulses, respectively. The pumping fluences were 21 μJ cm⁻² and 130 μJ cm⁻². **b** Normalized spectra at fixed white light delay $t_{WL} = 700$ fs of the maps in **c** and **e**. A Gaussian smoothing ($\sigma = 1$) has been applied to both traces. **d, f** Normalized pump-probe traces of the maps in **c** and **e** integrated over the region (1.4–1.7 eV) and (1.5–1.8 eV), respectively.

Our broadband measurements, however, put these interpretations in a new perspective and show that the early-time photo-induced transparency is not a universal feature of the ultrafast response of bulk BP, but that it vanishes for $E_{pr} > 2$ eV. This is an indication that the Pauli blockade effect induced by above-gap photo-excitations targets only the lowest-energy optical band associated to the dispersion along the stacking direction. To clarify this point, we compare in Fig. 2a the DFT optical absorption computed including in the calculation all the optical transitions (black line), and only the transitions between the last valence band (lVB) and the first conduction band (fCB) (yellow-

shadowed area). The two curves overlap up to ∼1.7 eV and then start to deviate, until the contribution of the optical transitions from lVB to fCB is almost suppressed. As transitions to other bands dominate above 2 eV, the probe can be absorbed even if the lVB and fCB are already occupied by the photo-excited carrier population, thereby overcoming the Pauli blocking effect. We stress that this effect is not related to a specific pump photon energy and photo-excitation by pumps with smaller photon energy (0.65 eV) but still larger than the band gap, induces a similar spectral response (Supplementary Fig. 4).

The broadband photo-induced transparency below 2 eV is not present when the sample is excited by pulses with photon energy smaller than the MIR gap (Fig. 2e). Strikingly, the spectral response with long-wavelength pumps displays a qualitatively different response, that is characterized by a negative feature peaked at ∼1.7 eV (Fig. 2b) and exhibits a long-lived dynamics (Supplementary Fig. 3). The suppression of the Pauli blocking effect is an indication that no optical transitions occur along the stacking direction upon the photo-excitation. Moreover, the emergence of a transient response that is well localized in frequency is in stark contrast with the calculated optical absorption of bulk BP (Fig. 2a), that does not display any particular resonance at 1.7 eV. Based on these arguments and the assignments in refs. [36,41,49], we tentatively identify this signal with the appearance of a transient absorption from the lowest-energy exciton resonance in the monolayer phosphorene, whose energy exactly matches the observed MIR-induced signal (Fig. 2a, green curve). In contrast to previous transient absorption measurements in few-layer BP that reported a derivative-like signal at the exciton resonance[18], our measurements display a signal centered at the expected $E_{11}$ transition, with no significant energy shifts within the time window considered in Fig. 2e. This may be an indication that, at least at early-times, many-body interactions (such as renormalization of the band gap and exciton binding energy, which would determine a pump-induced energy shift in the data) are negligible. A blue-shift of the resonance is found for $t_{WL}$ >15 ps (Supplementary Fig. 3) and attributed to carrier-phonon interactions, as already observed in other layered semiconductors[50,51].

**Characterization of the MIR-induced resonance.** In order to shed light on the mechanism leading to the observed resonance, we studied its amplitude at different pump photon energy across the band gap of bulk BP. As detailed in Supplementary Note 9, we quantified the intensity of the transient signal associated to the resonance at each pump photon energy. We plot in Fig. 3a the results of this analysis for two different fluence regimes of the MIR pump pulse. We observe that the signal is quenched at small pump photon energies. In particular, the resonance is suppressed at ~200 meV at low MIR fluences, while the cut-off edge is red-shifted at higher pumping fluences. We associate this cut-off to the effective band gap energy of the material upon the MIR photo-excitation, which is found to trigger a transient gap closure. While it would be tempting to ascribe the red-shift of the response to a carrier-induced band gap renormalization, as observed in ref. [22], we stress that the photo-injected excess energy here is smaller than the equilibrium band gap and no above-gap free-carrier population is excited in the linear response. In order to rule out population effects due to two-photon absorption, we studied the pump fluence-dependence of the response to MIR excitation. As evidenced in Supplementary Fig. 7, the signal associated to the exciton resonance does not scale as the square of the fluence (as it would be expected if two-photon absorption processes were involved), but it is consistent with a coherent effect that scales with the amplitude of the electric field.

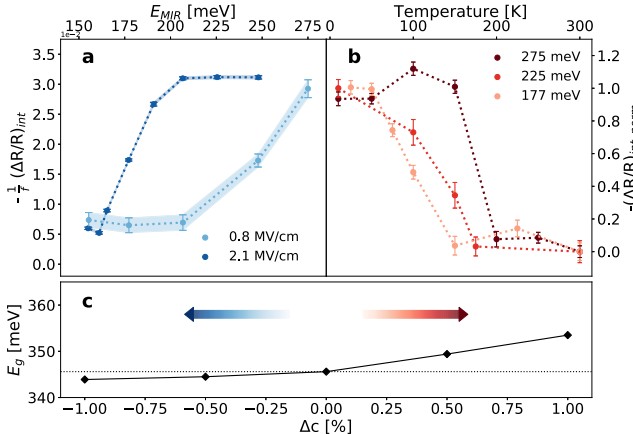

**Fig. 3 Non-adiabatic control of the black phosphorus gap with mid-infrared (MIR) pulses. a** Intensity of the MIR-driven resonance at 10 K as function of the photon energy of the MIR pump in a low ($f = 130 \, \mu J \, cm^{-2}$, light blue line) and high ($f = 890 \, \mu J \, cm^{-2}$, dark blue line) field strength regime. Each point in the plot is the result of an integration over a specific probe energy- and time-range (1.4–1.8 eV and 0–750 fs, respectively). The curves are divided by the corresponding fluence. **b** Normalized intensity of the resonance as function of the sample temperature for three different photon energies of the MIR pump. Error bars in **a**, **b** indicate the standard deviation associated to the energy- and time-integration. **c** DFT-calculated band gap energy ($E_g$) in bulk BP with modified interlayer distance (**c**). Negative (positive) values of Δc correspond to a compression (expansion) along the stacking direction.

While the further red-shift of the band gap at higher MIR fluence (Fig. 3a) could suggest a thermally driven effect, we ruled out this possibility by carrying out temperature-dependent measurements at fixed MIR pump photon energy. In Fig. 3b, each point indicates the intensity of the resonance measured at a given temperature and pump photon energy (details in Supplementary Note 10). For all the three examined MIR photon energies, the resonance disappears at high temperature, hence providing a clear indication that its emergence is related to the equilibrium band gap energy. Importantly, the high temperature cut-off for the dynamical resonance signal occurs at higher temperature when the MIR photon energy is larger. The temperature-dependence of the gap energy is then consistent with the equilibrium thermoelectric properties of BP, but follows an opposite trend with respect to the MIR fluence-dependent measurements in Fig. 3a.

In order to further rule out thermal effects, we carried out double-pumped pump-probe experiments employing a three-pulse scheme. As detailed in Supplementary Note 13, we used a differential acquisition to measure the MIR-driven optical response of BP after a preceding photo-excitation with visible pump pulses. When free photo-carriers are excited by the high-energy pump, the resonance in the MIR transient spectrum is suppressed and a broadband photo-bleaching dominates the response (Supplementary Fig. 12f). This evidence indicates that, whenever absorption occurs (being from the ground or the excited state), the photo-injected carriers screen the resonance and modulate the optical response through Pauli blocking. This is a further confirmation that the observed resonance is related to a coherent non-adiabatic effect, rather than a population one.

## Discussion

The experimental observation of a MIR-driven narrow resonance at 1.7 eV in BP constitutes the major discovery of this report. This is rather unexpected in the equilibrium response function of the

bulk material, which is flat and featureless in that spectral region. Such a dramatic change in the response function exactly in correspondence of the energy of the single-layer exciton, suggests that the interlayer interactions are impulsively perturbed and thus the 3D screening environment is modified.

The nonlinear response to sub-gap excitation indicates that the transient emergence of a visible MIR-driven resonance is accompanied by a non-thermal closure of the infrared band gap (Fig. 3a). To identify a mechanism that could rationalize our experimental evidence, we ran a set of simulations with modified interlayer spacing ($c$) in bulk BP at equilibrium. The rationale of this approach is that the 3D screening (along with the corresponding band structure and gap energy) at equilibrium is ultimately set by the interlayer distance in the simulated material. By freezing the in-plane atomic distances, we calculated the band gap energy when the crystalline structure is compressed ($\Delta c < 0$) or strained ($\Delta c > 0$) along the stacking direction up to 1% (Fig. 3c). We found that when the interlayer distance is increased, the gap energy increases as well, in agreement with previous calculations[5]. This scenario is qualitatively analogous to an adiabatic thermal expansion, which, as observed both at equilibrium (Supplementary Fig. 1) and in temperature-dependent pump-probe measurements (Fig. 3b), is related to an enlargement of the infrared optical gap. The observed MIR-driven collapse of the band gap (Fig. 3a) is instead compatible with a contraction along the stacking direction.

We stress, however, that such comparison is only qualitative, while quantitatively the observed dynamical collapse of the infrared gap remains unexplained and cannot be rationalized solely as an adiabatic compression of the lattice. On the one hand, the calculated gap energy is much larger than the one extracted by the pump wavelength-dependent experiments. In this regard, it should be noted that in the simulated adiabatic compression the in-plane lattice parameters have been kept fixed, while further in-plane contractions could result in a larger band gap closure[52], as detailed in Supplementary Note 12 where hydrodynamic pressure-dependent simulations are discussed. On the other hand, the most relevant aspect distinguishing the observed MIR-driven response from an adiabatic contraction of the $c$-axis is the fact that the latter is expected to increase the screening of excitonic resonances, while the optical measurements revealed the transient emergence of a resonance which is instead compatible with a dynamical reduction of screening.

Based on the evidence collected so far, we speculate that the sudden reduction of the 3D screening that possibly unveils the single-layer exciton in our measurements could be related to a non-adiabatic coherent redistribution of the charge carriers driven by MIR pulses. In a simple picture, the bulk material at equilibrium can be seen as an electron bath in which the single-layer excitons are largely screened due to the high carrier mobility. Upon photo-excitation by MIR pulses, the material interacts with a long-wavelength in-plane electric field, which forces the electrons to a coherent motion, resulting in light-driven ac-currents. Due to the sudden charge redistribution, the electron bath is then less effective in screening the exciton, whose transient absorption is then detected by the broadband probe. Moreover, the establishment of such currents might also affect the layered structure of the material. Analogously to the magnetic attraction experienced by wires carrying homodirectional dc-current (Biot-Savart law), the driven in-plane transport of charge carriers may result in a contraction along the stacking direction, that, based on our DFT simulations, could justify the observed band gap closure. In this regard, investigating the coherent response triggered by intense THz fields might be decisive to assess the role of light-driven ac-currents in a layered semiconductor prototype such as BP.

Finally, we stress that it cannot be excluded that other processes, not directly involving the single-layer excitonic structures, may give rise to a similar response. For example, a peaked response may arise from photo-induced perturbation of indirect band gaps. However, this seems a more unlikely scenario as significant structural changes would be required to justify such an impulsive modification of the material band structure. Moreover, motivated by our experimental evidence, we have considered that the emergence of the MIR-driven resonance and the non-thermal collapse of the gap are related phenomena. If we ease this assumption, other pictures could be considered. While the gap collapse might be related to a bulk effect, the resonance peaked at the single-layer exciton could be an indication of a photo-induced expansion of the material, eventually leading to a transient structural detachment of the surface layers. We deem this scenario unlikely for two reasons: (1) the deformation needed to completely decouple the external layers would be massive and yet reversible; (2) the response is peculiar for sub-gap excitation, while energy absorption is far larger for high-photon energy. Nevertheless, time-resolved structural determination (X-ray or electrons diffraction) would be needed to unequivocally rule out such effects.

In summary, we have studied the non-equilibrium dielectric environment in bulk BP upon photo-excitation by energy-tunable ultrashort pulses. Our ab initio DFT calculations have shown that the optical absorption below 2 eV is dominated by transitions involving energy bands dispersing along the stacking direction, making this spectral region intrinsically linked to the enhanced 3D screening in the bulk material. We have found that high-photon energy excitations uniformly target this whole band, leading to a spectrally flat photo-induced transparency through Pauli blocking. This bulk-like spectral response is suppressed when the sample is photo-excited by sub-gap MIR pulses, which reveal instead a peaked transient response, that matches the lowest-energy exciton resonance in the monolayer phosphorene. By unveiling how low energy ac-currents may modify the screening of excitonic resonances in quantum materials, our findings potentially enable an ultrafast control of screening in layered semiconductors, which is of paramount importance for 2D materials-based optoelectronic applications.

## Methods
**Pump-probe experiments**. We performed broadband non-equilibrium reflectivity measurements on bulk BP at different temperatures (10 K–300 K). The experimental apparatus was pumped by a Pharos Laser (by Light Conversion), which delivers 400 μJ pulses with 1.2 eV photon energy. We photo-excited the sample by tunable, carrier-envelope phase stable MIR pulses (155–275 meV) obtained by difference frequency generation in a GaSe crystal of two near-infrared pulses, both generated by a Twin Optical Parametric Amplifier (Orpheus TWIN by Light Conversion). The wavelength of the MIR pulses was measured by a home-built Michelson interferometer, equipped with a mercury cadmium telluride detector. The 3.1 eV-pump was obtained by second harmonic generation in a β-barium borate crystal of the output of a Non-Collinear Parametric Amplifier (Orpheus-N by Light Conversion). Out-of-equilibrium reflectivity was probed by a supercontinuum white-light extending from 1.3 eV to 2.2 eV generated by self-phase modulation of the Pharos output in a sapphire crystal. The reflected probe beam was diffracted, and its frequency components were acquired by a linear array of silicon photodiodes (NMOS by Hamamatsu) synchronized with the laser repetition rate. The signal-to-noise ratio was increased by the simultaneous acquisition and division by a reference beam (not interacting with the sample) to suppress the white-light fluctuations. The laser repetition rate was set to 5 kHz. We report that higher repetition rates induce the onset of long-lasting thermal signals that obscure the fast dynamics. The probe was polarized along the armchair direction ($\hat{x}$), while both the visible and the MIR pumps were cross-polarized. The sample axes were identified by polarization-dependent visible pump-probe measurements, according to ref.[16]. The FWHM of the probe beam was 30 μm, small enough to ensure a uniform illumination of the irregular sample surface. No physical correction of the temporal chirp of the broadband white-light probe was performed, but all the data were post-processed to compensate for the dispersion. Samples were provided by HQ graphene and glued on an oxygen-free copper substrate. All the experiments were performed on freshly cleaved multi-layer BP by in situ mechanical exfoliation.

Samples were kept under vacuum ($<10^{-8}$ mbar) for the entire duration of the experiment to prevent air degradation and cooled down to cryogenic temperatures using a closed-cycle liquid helium cryostat.

**Ab initio calculations.** The ground state structural and electronic properties of bulk BP have been evaluated at the DFT level, using the Quantum Espresso package[53].

The structural relaxation has been performed using the PBE[54] approximation for the exchange correlation potential together with the plane wave basis set, while ONCVPSP[55] norm-conserving pseudopotentials have been adopted to model the electron–ion interaction; the kinetic energy cut-off for the wave-functions has been fixed to 90 Ry, while the Brillouin zone has been sampled with a $16 \times 16 \times 8$ k-point grid. vdW interactions between layers has been included using the Grimme-D2 parametrization[56].

Both atomic positions and lattice parameters (a conventional orthogonal unit cell has been used, with eight atoms per cell) were relaxed up to when forces acting on each atom were below $3 \times 10^{-4}\,\mathrm{eV\AA^{-1}}$. The obtained lattice parameters $a = 4.43$ Å, $b = 3.33$ Å, $c = 10.48$ Å are in good agreement with experimental[57] and theoretical results[58].

Hybrid-DFT (using Gau-PBE hybrid functional[59]) has been used to perform electronic band structure calculations. The equilibrium charge density and electronic Kohn-Sham states have been computed using a $20 \times 20 \times 10$ k-point grid to sample the Brillouin zone, in combination with a $4 \times 4 \times 2$ grid for the sampling of the Fock operator. The obtained direct electronic band gap of 0.34 eV (located at the $\Gamma$ point) turns out to be in reasonable agreement with the experimental one[60].

Optical properties have been evaluated using the Yambo code[61], at the independent-particle level. The dielectric function has been calculated as:

$$\varepsilon_\alpha(E) = 1 + \frac{16\pi}{\Omega}\sum_{c,v}\sum_k \frac{1}{E_{ck}-E_{vk}}\frac{|r_{vc}^\alpha|^2}{(E_{ck}-E_{vk})^2-(E+i\gamma)^2} \tag{1}$$

where $E_{ck}$ ($E_{vk}$) are the conduction (valence) electronic states computed at the Gau-PBE level, $r_{vc}^\alpha$ are the interband electric dipoles between a valence and a conduction state projected along direction $\alpha$ (the direction of light polarization) and computed using Covariant approach[61]; finally, $\gamma$ is a broadening, here always fixed to 0.1 eV.

## Data availability

The data that support the findings of this study are available from the corresponding author upon reasonable request.

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

## Acknowledgements

This work was supported by the MIUR through the PRIN program No. 2017BZPKSZ and by the European Commission through the projects INCEPT (ERC-2015-STG, Grant No. 677488) and COBRAS (ERC-2019-PoC, Grant No. 860365).

## Author contributions

D.F. conceived the experiment and managed the project. A.M. and F. Giusti designed the experiment and performed the time-resolved measurements. F. Glerean, G.J., E.M.R., and S.Y.M. contributed to the experimental work. M.Z., D.V., M.R., and E.M. performed the DFT calculations. P.D.P. and A.P. provided the sample and performed the FTIR measurements. A.M. and D.F. wrote the manuscript, with contributions from all the authors.

## Competing interests

The authors declare no competing interests.
