## [Peer Review File · Nature Communications]

Reviewers' Comments:

Reviewer #1:

Remarks to the Author:

Montanaro et al. describe in their manuscript infrared-pump (broadband) visible-probe experiments on thin layers of black phosphorus. The pump wavelength for the experiments is either above or below the bandgap, resulting in a significant change of the observed transient reflection. While the response after high energetic excitation is attributed to Pauli blocking, the response to the low energetic photons is attributed to an excitonic response. To further analyze the results, the authors modelled the optical properties of the samples via DFT modelling.

The assumptions and conclusions drawn by the authors are well justified within the manuscript, which is in general well written and the relevant references are discussed. The overall quality of the figures is very high, also the content of the supporting material is very good and helpful to understand the background of the measurements.

The quality of the data and the novelty of the findings justify publication in Nature Communications, which I fully support. I only have four minor points that the authors might revise:

- The subfigures have different fonts for the numbering (e.g. (A) in Figs. 1 & 2)
- Subfigure 1 A could be more clear as it describes the core of the manuscript. I suggest to improve it to more clearly convey the message of the manuscript.
- The temperature dependence could be added to the main manuscript as I find it too interesting for the supporting material.
- The formatting of the references is not always correct: Refs. 7 and 10 start with "

Reviewer #2:

Remarks to the Author:

The authors investigate the transient optical properties of bulk BP by pump-probe spectroscopy, with assistance of DFT calculations. Since BP is an infrared layered material with widely tunable direct bandgaps and attracts increasingly attention in recent years, this topic itself is very interesting, and will be highly welcome by the science community of 2D materials and low-dimensional physics.

In the experiments, pump energies of 3.1 eV (above the bandgap) and 0.275 eV (below the bandgap) are both employed, and the induced changes in the transient absorption of bulk BP are probed by broadband reflectivity measurements in the range of 1.34-2.1 eV. For the above bandgap pump, the authors observed the phenomenon of photo-bleaching and photo-induced absorption, in agreements with previous studies [17, 18, 20].

The main contribution of this manuscript is the study of below bandgap pump. The authors claim that they observed single-layer exciton resonances in bulk BP, which is induced by the non-thermal suppression of the 3D dielectric screening of excitons upon the pump. To me, the conclusion is not convincing at all, and the analysis and claims in the manuscript are quite confusing.

Below are the details,

1. The authors mentioned that "Although distinct higher-order transitions are theoretically predicted in the single-layer limit [4], the most prominent resonance is the lowest-energy exciton (E11), ...".

This statement is confusing and ambiguous. In monolayer BP, there are no subbands because of lack of interlayer interactions. Thus, only E11 transition is allowed, higher-order (E22, E33, ...) transitions do not exist intrinsically. Ref. [4] predicts that 2s, 3s, ... exciton states may contribute to the optical absorption, and the 1s resonance dominates. In fact, these exciton states are all related to the E11 transition.

2. The authors claimed that "Importantly, as the layer number (L) is increased, the E11 resonance (along with the band gap) monotonically shifts to lower energies, as a consequence of the enhanced screening of the electron-hole Coulomb attraction."

As is well-known, the E11 resonance energy (or the bandgap) decreases with increasing layer number, due to strong interlayer interactions, instead of the screening of electron-hole interactions.

3. The authors claimed that "In particular, the optical transitions within 0.34 eV (gap energy in bulk BP) and 2 eV (gap energy in the monolayer) are intrinsically related to the three-dimensional nature of the material [37]"

This statement is quite confusing. We know that monolayer BP is a 2D material, the electrons in which are confined in the 2D layer plane, so are the holes. The band dispersion in monolayer BP is intrinsically 2D, whose density of states are markedly different from those in 3D counterparts. Why the authors say the energy gap in monolayer is intrinsically related to the three-dimensional nature of the material [37]?

4. As seen in Fig. 2A, the authors say "As transitions to other bands dominate above 2 eV, ..."
What are the "other bands", besides the last valence band (IVB) to the first conduction band (fCB)? Does the authors mean that "other bands" are related to the transition with ZZ polarizations? If that, previous calculations (doi:10.1088/2053-1583/2/4/044014) predict that the transition energy is around 3.8 eV, far larger than 2.0 eV.

5. The most confusing is the author's claim that they observed single-layer exciton resonances (1.7 eV) in bulk BP under below bandgap pump (0.275 eV). Unfortunately, this is the major conclusion of the paper. I can not find any evidence from the data to support this conclusion. As is well known, in 3D materials, the exciton is 3D in nature. Even in few-layer case, the electron and hole, constituting an exciton, are not confined in a single layer. The exciton wavefunction diffuses throughout the whole material in the out-of-plane direction (DOI: 10.1103/PhysRevB.89.235319). Hence, 2D exciton is not expected to be existed in bulk BP. I believe that the major claim in the paper is invalid and the paper doesn't meet the high standard of the journal.

REVIEWER COMMENTS

Reviewer #1 (Remarks to the Author):

Montanaro et al. describe in their manuscript infrared-pump (broadband) visible-probe experiments on thin layers of black phosphorus. The pump wavelength for the experiments is either above or below the bandgap, resulting in a significant change of the observed transient reflection. While the response after high energetic excitation is attributed to Pauli blocking, the response to the low energetic photons is attributed to an excitonic response. To further analyze the results, the authors modelled the optical properties of the samples via DFT modelling.

The assumptions and conclusions drawn by the authors are well justified within the manuscript, which is in general well written and the relevant references are discussed. The overall quality of the figures is very high, also the content of the supporting material is very good and helpful to understand the background of the measurements.

The quality of the data and the novelty of the findings justify publication in Nature Communications, which I fully support. I only have four minor points that the authors might revise:

We thank the Reviewer for the work done, for their support to the publication of our manuscript, and for appreciating the novelty and relevance of our work. We address their points in the following.

- The subfigures have different fonts for the numbering (e.g. (A) in Figs. 1 & 2)

We thank the Reviewer for spotting this discrepancy, which we corrected in the revised version of the manuscript.

- Subfigure 1 A could be more clear as it describes the core of the manuscript. I suggest to improve it to more clearly convey the message of the manuscript.

We thank the Reviewer for this feedback. Following their suggestions, we modified it to include the white light probe pulse and its broadband detection, to explain more clearly the experimental conditions of the work. The new version of Figure 1 is reported above, along with its revised caption.

Figure 1: Optical fingerprint of 3D screening in bulk black phosphorus. (A) Sketch of the broadband reflectivity measurements carried out on bulk BP following above- (blue) and below-gap (red) photo-excitation. The transient screening suppression upon mid-infrared photo-excitation reveals the undressed exciton in the monolayer phosphorene. BP crystal structure was adapted from ref. [24]. (B) Simplified sketch of the electronic structure at the Γ point, as indicated in the first Brillouin zone. The arrows represent the high-photon energy (blue) and sub-gap (red) photo-excitation. (C) Calculated imaginary part of the dielectric function in bulk BP along the armchair (\hat{x} , blue) and zigzag (\hat{y} , grey) direction, convoluted with a numerical broadening of 100 meV. The blue peaks denote the lowest-energy exciton resonances (E_{11}) in monolayer (1L), bilayer (2L), trilayer (3L) etc. BP, adapted from ref. [30].

In the same fashion, we slightly modified also Figure 2 (see below) to highlight more clearly the different spectral response to above- and sub-gap photoexcitation, and changed the caption accordingly. In particular, in Figure 2A we included the calculated optical absorption also below 1.38 eV to show the electronic gap and increased the spectral weight of the exciton resonance with respect to the bulk response. In Figure 1B, the spectra are now cuts at fixed $t_{WL}=700$ fs (instead of $t_{WL}=500$ fs) to more clearly emphasize the measured exciton signal.

Figure 2: Above- and below-gap photo-excitation in bulk black phosphorus. (A) Normalized DFT-calculated total optical absorption of bulk BP (thick black line). The yellow area indicates the optical absorption obtained when only the transitions from the last valence band (IVB) to the first conduction band (fCB) are included in the calculation. The green Gaussian shape indicates the lowest-energy exciton resonance in the single-layer limit. The energy scale is uniformly spaced from 1.34 eV up to 2.1 eV (probe energy window), and it is shrunk below 1.38 eV and above 2.1 eV. (C),(E) Transient reflectivity map measured on bulk BP at 10 K upon photo-excitation by high-photon energy ($3.1 \text{ eV} > E_g$) and sub-gap ($275 \text{ meV} < E_g$) pulses, respectively. The pumping fluences were $21 \mu\text{J cm}^{-2}$ and $130 \mu\text{J cm}^{-2}$. (B) Normalized spectra at fixed $t_{WL}=700$ fs of the maps in (C) and (E). A Gaussian smoothing ($\sigma=1$) has been applied to both traces. (D),(F) Normalized pump-probe traces of the maps in (C) and (E) integrated over the region (1.4-1.7 eV) and (1.5-1.8 eV), respectively.

-The temperature dependence could be added to the main manuscript as I find it too interesting for the supporting material.

We thank the Reviewer for appreciating the importance of the temperature dependent measurements that we carried out. The purpose of this set of measurements was to highlight how differently the sample temperature (which in turn modifies the gap energy) affects the optical response to high- and low-photon energy excitations. When above-gap pulses are used (both in the visible and in the near-infrared range), no appreciable temperature dependence is detected (Section S5 in Supplementary Materials). Conversely, the optical signature of the exciton resonance is closely related to the sample temperature, and vanishes when, as a consequence of the increased temperature, the band gap largely exceeds the photon energy of the mid-infrared excitation.

Although we agree with the Reviewer that these are indeed new results in literature, for the sake of clarity, we would prefer not to include a detailed description of the temperature dependent measurements and their analysis (Secs. S5&S10 in Supplementary Materials) in the main manuscript, while keeping only the temperature-dependent emergence of the single-layer exciton at different mid-infrared photon energies, as already reported in Fig. 3B. We reckon that this remains the most important result and the only one which is essential to substantiate the main conclusions of the paper.

- The formatting of the references is not always correct: Refs. 7 and 10 start with "

We thank the Reviewer for their careful reading. We formatted the references pointed out by the Reviewer, along with Refs. 53 and 54, which had the same issue. We also deleted Refs. 1-2 that were cited only in the abstract and shifted the reference numbering accordingly.

Reviewer #2 (Remarks to the Author):

The authors investigate the transient optical properties of bulk BP by pump-probe spectroscopy, with assistance of DFT calculations. Since BP is an infrared layered material with widely tunable direct bandgaps and attracts increasingly attention in recent years, this topic itself is very interesting, and will be highly welcome by the science community of 2D materials and low-dimensional physics.

In the experiments, pump energies of 3.1 eV (above the bandgap) and 0.275 eV (below the bandgap) are both employed, and the induced changes in the transient absorption of bulk BP are probed by broadband reflectivity measurements in the range of 1.34-2.1 eV. For the above bandgap pump, the authors observed the phenomenon of photo-bleaching and photo-induced absorption, in agreements with previous studies [17, 18, 20]. The main contribution of this manuscript is the study of below bandgap pump. The authors claim that they observed single-layer exciton resonances in bulk BP, which is induced by the non-thermal suppression of the 3D dielectric screening of excitons upon the pump. To me, the conclusion is not convincing at all, and the analysis and claims in the manuscript are quite confusing.

GENERAL REPLY: We thank Reviewer #2 for their effort and their comments. We appreciate that Reviewer #2 recognizes the novelty of our work. In particular, the study of the response of bulk BP to a long-wavelength excitation was missing in literature. The main observation that a photo-excitation at long-wavelength gives rise to a qualitatively different response with respect to photo-excitation with high photon energy pulses will possibly have a high impact on the 2D community, as also stated in the report of Reviewer #1.

With regards to the objections of Reviewer #2 to our main conclusion, we ascertain that the main point of the manuscript is the report of the experimental evidence that the response of bulk BP to long-wavelength photo-excitations is anomalous. We will argue in the following that Reviewer #2's objection to our results may mostly be related to minor points regarding the theoretical discussion of the data that do not jeopardize the robustness of our experimental evidence.

We argue in the following the reasoning that led to our interpretation of the data, add new data on double-pumped pump-probe experiments, which further confirm our interpretation, and detail the changes made to the manuscript to clarify the roots of our interpretation.

DETAILED REPLY: As correctly outlined by Reviewer #2, our paper relies on the comparison of the broadband transient optical response of bulk BP to photo-excitation above- and below-gap. Conversely to the statement of the Reviewer, we did not study the transient reflectivity at 2 pump wavelengths but rather the detailed dependence of the response to a tuning of the pump photon energy (pump energy used 3.1; 0.83; 0.71; 0.65; 0.275; 0.248; 0.207; 0.177; 0.155 eV).

As Reviewer #2 mentions, visible pump-probe experiments on BP are widely discussed in literature and our data confirm a well-known photo-bleaching signal due to Pauli blocking effect.

The main discovery reported in the manuscript, however, is that the response to sub-gap mid-infrared pumps is *markedly different*. While photo-exciting BP with high-photon energy pumps results in a nearly wavelength-independent change in reflectivity, photoexcitation by sub-gap mid-infrared pumps results instead in a response which is well-localized in energy (1.7 eV), which matches the resonance of the exciton in single layer phosphorene. The experimental evidence for this is very clear and its dependence on the pump wavelength has been thoroughly characterized in our manuscript.

The tentative explanation that we propose, albeit justified only at a qualitative level, is based on several arguments that we discuss throughout the manuscript and summarize here. The lack of a broadband photo-bleaching signal and the absence of a clear resonance at 1.7 eV in the equilibrium absorption spectrum of bulk BP indicates that the mid-infrared photoexcitation triggers a modification of the response function of the bulk material. Moreover, the strong, yet opposite, dependence of the signal on both the photon energy of the mid-infrared pump employed and the sample temperature indicate that the transient appearance of the resonance can neither be attributed to a population nor to a thermal effect. We also excluded two-photon absorption processes by studying the linearity of the response.

To give further evidence of the validity of our interpretation, we have inserted in the revised version of the Supplementary Materials (Note 13) additional data acquired by a three-pulse technique, which show that the MIR-driven exciton resonance is suppressed when free carriers are previously photo-injected in the sample by above-gap pumping. In these conditions, the response to MIR fields is dominated by a broadband photo-bleaching. This is a further confirmation that, whenever above-gap absorption occurs, the exciton resonance is screened and population effects modulate the optical response through Pauli blocking.

Altogether, we concluded that the most plausible explanation of the effect reported is that whatever masks in the bulk optical conductivity the single-layer response at equilibrium is dynamically “contrasted” by long-wavelength mid-IR photo-excitation. The merit of our manuscript is the experimental observation of such an effect. We stress that the exact mechanism leading to the emergence of a 2D response function in bulk BP subsequently to mid-IR excitation remains an open question and we believe our manuscript will stimulate further research in this direction.

Below are the details,

1. The authors mentioned that “Although distinct higher-order transitions are theoretically predicted in the single-layer limit [4], the most prominent resonance is the lowest-energy exciton (E_{11}), ...”.

This statement is confusing and ambiguous. In monolayer BP, there are no subbands because of lack of interlayer interactions. Thus, only E_{11} transition is allowed, higher-order (E_{22} , E_{33} , ...) transitions do not exist intrinsically. Ref. [4] predicts that $2s$, $3s$, ... exciton states may contribute to the optical absorption, and the $1s$ resonance dominates. In fact, these exciton states are all related to the E_{11} transition.

We understand that this sentence may be confusing, and we thank the Reviewer for pointing this out. We changed the original sentence to the following one: “Although higher-energy exciton states are theoretically predicted in the single-layer limit [5*], the most prominent one is the lowest-energy $1s$ resonance of the E_{11} exciton transition, ...” that, we agree with Reviewer #2, is clearer. However, we stress that a detailed discussion of the optical absorption of the phosphorene monolayer (may be due to the hydrogen-like series of the E_{11} transition or higher-order ones) is beyond the scope of this work and the only important point that we raise here is that there is no optical absorption in the single-layer below 1.7 eV.

* The reference numbering has been modified in the revised version of the manuscript, so Ref. [4] has now become Ref. [5].

2. The authors claimed that “Importantly, as the layer number (L) is increased, the E_{11} resonance (along with the band gap) monotonically shifts to lower energies, as a consequence of the enhanced screening of the electron-hole Coulomb attraction.”

As is well-known, the E_{11} resonance energy (or the bandgap) decreases with increasing layer number, due to strong interlayer interactions, instead of the screening of electron-hole interactions.

We thank Reviewer #2 for this comment. We agree with him/her that our statement may not be appropriate. The aspect which we wanted to emphasize with this sentence was not the 3D screening of the Coulomb attraction, but solely the observation of the redshift of the gap of the material as thickness is increased, without addressing the specific microscopic mechanism leading to this.

We changed the sentence to clarify this “Importantly, as the layer number (L) is increased, the E_{11} resonance (along with the band gap) monotonically shifts to lower energies, as a consequence of the strong interlayer interactions.”, as they suggested.

We acknowledge that the redshift of the band gap at progressively higher layer number is more directly caused by the interlayer interactions, while the screening of the electron-hole Coulomb attraction is an accompanying effect that,

although present, mainly affects the exciton binding energy and hence the E_{11} exciton resonance as shown by Many Body Perturbation Theory calculation in Fig.2 of Ref. [5] (Tran et al., 2014).

However, we would like to emphasize that this detail, that we are glad to have clarified in the revised manuscript, does not compromise neither the reliability of our experimental results, nor the validity of the conclusions that we draw. As a matter of fact, while the monolayer phosphorene exhibits a narrow excitonic resonance at 1.7 eV, the optical absorption of the bulk material is nearly wavelength-independent down to 0.3 eV, where a mid-infrared electronic gap opens. Once again, the relevant aspects are i) the absorption in bulk BP is flat in the visible region and no narrow resonances are observed and ii) the absorption in the visible and near-IR is determined by the 3D nature of the material.

The experimental evidence shows instead that photo-excitation in the visible gives a transient response broad in frequency, while sub-gap excitation reveals a transient response peaked at 1.7eV. As matter of fact, the observation that sub-gap excitation drives the emergence of a resonance in the bulk material at the same energy of the single layer exciton, may indirectly confirm that indeed coulomb attraction is not strongly affected by the 3D nature of the material.

3. The authors claimed that “In particular, the optical transitions within 0.34 eV (gap energy in bulk BP) and 2 eV (gap energy in the monolayer) are intrinsically related to the three-dimensional nature of the material [37]” This statement is quite confusing. We know that monolayer BP is a 2D material, the electrons in which are confined in the 2D layer plane, so are the holes. The band dispersion in monolayer BP is intrinsically 2D, whose density of states are markedly different from those in 3D counterparts. Why the authors say the energy gap in monolayer is intrinsically related to the three-dimensional nature of the material [37]?

We think that Reviewer #2 misread our sentence. In the sentence quoted we are not stating what the Reviewer #2 says in their comment: “*the energy gap in monolayer is intrinsically related to the three-dimensional nature of the material [37]*”. The sentence in the manuscript that the Reviewer is questioning refers uniquely to the optical properties of the bulk material (in our paper we discussed and study only bulk material) and claims that all the optical transitions that occur within a certain energy range ([0.34 eV, 2 eV], specifically) are related to its three-dimensional nature. This is well known in literature and discussed extensively in Section 11 of the Supplementary Materials of our manuscript, where we analyse the DFT calculated optical absorption of bulk BP for light polarized along the armchair (\hat{x}) direction (blue curve in Fig.1C). By identifying the dipole matrix elements with the highest dipole strength within specific energy ranges, we were able to identify which regions of the Brillouin zone contribute the most to a given portion of the absorption spectrum. This analysis reveals that from ~ 0.3 eV up to ~ 2 eV the optical absorption is dominated by transitions from the last valence band to the first conduction band, characterized by wave vectors with a progressively increasing k_z component. This means that the absorption in this range is mostly due to the dispersion along the stacking direction, which, as a matter of fact, is intrinsically related to the three-dimensional nature of the material.

Needless to say that we agree with the Reviewer that the band dispersion in the single-layer phosphorene has an intrinsic 2D nature, and we are well aware that its density of states is completely different from the one of its bulk counterpart. Once again, our study focuses solely on bulk material.

4. As seen in Fig. 2A, the authors say “As transitions to other bands dominate above 2 eV, ...” What are the “other bands”, besides the last valence band (IVB) to the first conduction band (fCB)? Does the authors mean that “other bands” are related to the transition with ZZ polarizations? If that, previous calculations (doi:10.1088/2053-1583/2/4/044014) predict that the transition energy is around 3.8 eV, far larger than 2.0 eV.

As mentioned above, we discussed in detail the DFT calculated optical absorption in Section 11 of the Supplementary Materials. By evaluating the oscillator strength of the dipole matrix elements along the armchair axis, we found that up to 2 eV the optical absorption of bulk BP can be fully accounted for only by transitions between the last valence band (IVB) and the first conduction band (fCB). Instead, at higher energies (>2 eV), the most prominent contribution comes from transitions between the penultimate valence and the second unoccupied conduction band. We refer to these bands in the sentence reported by the Reviewer. This can be appreciated more clearly in Fig. 2A, where we compare the optical absorption obtained by including in the calculations all the available bands (thick black line) or just the IVB and the fCB (yellow-shaded area). Our theoretical findings are further confirmed by the broadband reflectivity measurements at above-gap pumping energies, which show a clear cut-off at ~ 2 eV, indicating a suppression of the

visibility of the Pauli blockade effect above this energy. Finally, we thank the Reviewer for pointing out the calculations in Ref. [doi:10.1088/2053-1583/2/4/044014, Tran et al., 2015], even though they describe only single-layer and few-layer BP and do not directly apply to our work. A citation to this paper (Ref. [34]) has been included in our manuscript only as a benchmark to highlight the difference between bulk BP and phosphorene.

5. The most confusing is the author's claim that they observed single-layer exciton resonances (1.7 eV) in bulk BP under below bandgap pump (0.275 eV). Unfortunately, this is the major conclusion of the paper. I can not find any evidence from the data to support this conclusion. As is well known, in 3D materials, the exciton is 3D in nature. Even in few-layer case, the electron and hole, constituting an exciton, are not confined in a single layer. The exciton wavefunction diffuses throughout the whole material in the out-of-plane direction (DOI: 10.1103/PhysRevB.89.235319). Hence, 2D exciton is not expected to be existed in bulk BP. I believe that the major claim in the paper is invalid and the paper doesn't meet the high standard of the journal.

We believe that the Reviewer's statement "*the major claim in the paper is invalid*" is based on theoretical calculations on the exciton wavefunction. While we agree with the theoretical statement of the Reviewer (i.e. in bulk BP excitons have a 3D wavefunction), we stress that the situation we are having in our experiment is significantly different. In our experiment, bulk BP is strongly driven by long wavelength AC-fields and it is not at all clear that the exciton wavefunctions obtained in literature calculations are a good starting point for describing the effect reported here.

So overall, we agree with the Reviewer that the observation of a resonance at the frequency of the single-layer exciton in the bulk material is, at very least, surprising. Nonetheless, the experimental evidence, which is the core of our work, seems to suggest the opposite. We find it hard to believe that the observation of the emergence of a narrow feature in the optical conductivity which is exactly at the energy of the exciton of the single layer is merely coincidental.

We have revised the manuscript aiming at clarifying the points which Reviewer #2 found confusing and the spirit of the paper. We added supporting evidence to our interpretation and hope that this responds to the criticisms of Reviewer #2, whom we thank again for their effort.

Reviewers' Comments:

Reviewer #1:

Remarks to the Author:

The authors addressed all points well and I fully support publication of the current manuscript in Nature Communications!

Reviewer #2:

Remarks to the Author:

I read the response of the authors and thanks for the effort. I agree with the authors that the observation of a resonance at 1.7eV is interesting. However, I'm still not convinced this is due to the monolayer resonance. The layer dependent band gap of few-layer BP is due to the strong inter-layer coupling, not the change of the screening. I'm wondering how a pump can eliminate the interlayer coupling within a bulk BP. The authors admit that the interpretation of the resonance is premature and full of speculation. I think the authors may need more intensive and full-scale simulations to get an idea, to extract the true reason of the resonance. Currently, this is still a puzzle.

REVIEWER COMMENTS

Reviewer #1 (Remarks to the Author):

The authors addressed all points well and I fully support publication of the current manuscript in Nature Communications!

We thank Reviewer#1 for the insightful comments and for supporting the publication of the paper in Nature Communications.

Reviewer #2 (Remarks to the Author):

I read the response of the authors and thanks for the effort. I agree with the authors that the observation of a resonance at 1.7eV is interesting. However, I'm still not convinced this is due to the monolayer resonance. The layer dependent band gap of few-layer BP is due to the strong inter-layer coupling, not the change of the screening. I'm wondering how a pump can eliminate the interlayer coupling within a bulk BP. The authors admit that the interpretation of the resonance is premature and full of speculation. I think the authors may need more intensive and full-scale simulations to get an idea, to extract the true reason of the resonance. Currently, this is still a puzzle.

We thank Reviewer#2 for reading our response and their comment.

We agree with the Reviewer that a full-scale simulation would be needed to confirm unequivocally the scenario that we propose. Unfortunately, the theoretical tools currently at our disposal are not suitable for such study. We stress, however, that the aim of the manuscript is to report the experimental discovery of this anomalous response and a full theoretical explanation of this effect is beyond the scope of the paper. The purpose of our manuscript is to stimulate the community by reporting this experimental finding.

We understand, thanks also to the comments of Reviewer#2, that our first version of the manuscript may have been misleading in this respect and worded strongly around the tentative explanation rather than focused on the description of the experimental evidence. We have revised the manuscript in order to clarify that the main merit of our work is the experimental observation that the optical response of BP to sub-gap photo-excitation is anomalous. We clearly emphasize in the revised manuscript that the explanation of the experimental evidence that we propose is tentative, albeit we still believe this is the most plausible one. For clarity, we outline other possible scenarios and explain why we deem them less likely representative of the leading physical mechanisms.

In detail, we changed the title of the paper in “*Anomalous non-equilibrium response in black phosphorus to sub-gap mid-infrared excitation*” to clearly state that the intention of the manuscript is to provide an experimental report, rather than a conclusive explanation of the effect.

We changed the abstract in order to highlight more clearly that the scenario proposed should be considered only as a tentative explanation of the experimental results.

We have modified several sentences throughout the text along the same line.

We have revised the Discussion section. In particular, we have included an initial paragraph to emphasize once again the primary experimental nature of the report. Furthermore, we have added a concluding paragraph to discuss alternative explanations to the peaked resonance observed. As explained in the text, perturbed indirect band gaps could trigger a transient absorption at selected energies. However, the necessary photo-induced changes in the material band structure would require substantial structural modifications, that are unlikely to take place. A photo-induced structural expansion, that would result in a sensitivity to just few layers, seems

equally improbable. Again, the interlayer spacing should increase enormously to justify the detachment of the surface layers.

Finally, we conclude that transient, and thus reversible, structural modifications are unrealistic explanations. A non-adiabatic redistribution of the charge carriers, accompanied by the subsequent modification of the dielectric environment, remains, in our opinion, the most plausible scenario.

We hope that this reply clarifies that the main focus of the work is the experimental report of the transient response of BP to mid-infrared excitation, that was missing in literature. We believe that the peculiar spectral response that we measured and characterized is, by itself, an interesting result for the broad scientific community and will motivate further research in this direction.